# Transgenerational Social Stress Alters Immune–Behavior Associations and the Response to Vaccination

**DOI:** 10.3390/brainsci7070089

**Published:** 2017-07-21

**Authors:** Alexandria Hicks-Nelson, Gillian Beamer, Kursat Gurel, Rachel Cooper, Benjamin C. Nephew

**Affiliations:** 1Department of Biomedical Sciences, Cummings School of Veterinary Medicine, Tufts University, North Grafton, MA 01536, USA; Alexandria.Hicks_Nelson@tufts.edu (A.H.-N.); kursatgurelcbu@gmail.com (K.G.); 2Department of Infectious Disease and Global Health, Cummings School of Veterinary Medicine, Tufts University, North Grafton, MA 01536, USA; Gillian.Beamer@tufts.edu (G.B.); Rachel.Cooper@tufts.edu (R.C.)

**Keywords:** social stress, transgenerational, IL-6, TNF, interferon, vaccination, social behavior, depression, anxiety

## Abstract

Similar to the multi-hit theory of schizophrenia, social behavior pathologies are mediated by multiple factors across generations, likely acting additively, synergistically, or antagonistically. Exposure to social adversity, especially during early life, has been proposed to induce depression symptoms through immune mediated mechanisms. Basal immune factors are altered in a variety of neurobehavioral models. In the current study, we assessed two aspects of a transgenerational chronic social stress (CSS) rat model and its effects on the immune system. First, we asked whether exposure of F0 dams and their F1 litters to CSS changes basal levels of IL-6, TNF, IFN-γ, and social behavior in CSS F1 female juvenile rats. Second, we asked whether the F2 generation could generate normal immunological responses following vaccination with *Mycobacterium bovis* Bacillus Calmette–Guérin (BCG). We report several changes in the associations between social behaviors and cytokines in the F1 juvenile offspring of the CSS model. It is suggested that changes in the immune–behavior relationships in F1 juveniles indicate the early stages of immune mediated disruption of social behavior that becomes more apparent in F1 dams and the F2 generation. We also report preliminary evidence of elevated IL-6 and impaired interferon-gamma responses in BCG-vaccinated F2 females. In conclusion, transgenerational social stress alters both immune–behavior associations and responses to vaccination. It is hypothesized that the effects of social stress may accumulate over generations through changes in the immune system, establishing the immune system as an effective preventative or treatment target for social behavior pathologies.

## 1. Introduction

The application of cutting edge techniques to animal models with limited ethological and/or translanational relevance is a limitation to the development of more effective treatments for social behavior disorders. The overall relevance of data on the biological basis of social behavior is directly dependent on the face, construct, and predictive validity of the animal model used. Despite this, most efforts towards improving best practices in social neuroscience are focused on technological advances. There is a clear need for a similar level of effort to be directed towards improving animal models, and the current study and previous related works suggest that the use of ethologically relevant transgenerational models will generate novel insights into the vast heterogeneity of disorders that affect social behavior, the growing importance of the immune system, and the potential for preventing social behavior disorders in future generations.

Integrated understandings of the etiologies, mechanisms and downstream generational consequences amongst disorders that affect social behavior such as depression, anxiety, autism, posttraumatic stress disorder (PTSD), bipolar disorder, and schizophrenia are unknown. Similar to the multi-hit theory of schizophrenia, social pathologies are mediated by multiple factors across generations, likely acting additively, synergistically, and/or antagonistically [1,2]. In human populations, a major challenge in studying the transgenerational development of social behavior psychopathology is that initiating events are difficult to identify and separate from transgenerational consequences. This area specifically benefits from the study of multigenerational rodent models, where initiation and consequences can be separated allowing identification of underlying mechanisms and designing appropriate interventions.

Exposure to social adversity, especially during early life, has been proposed to induce depression symptoms through immune mediated mechanisms [3,4]. Basal immune factors are altered in a variety of neurobehavioral models [5,6,7,8]. Our own studies using a multigenerational chronic social stress (CSS) rodent model of postpartum depression and anxiety [9,10,11,12,13,14,15,16,17] have also correlated transgenerational effects of social stress on F2 social behavior with changes in immune cytokines, in particular those involved in proinflammatory responses [18]. The CSS model involves exposing F0 lactating dams and their F1 litters to novel male intruder rats for an hour each day for 15 days during lactation (days 2–16). This stress induces depressed maternal care in the F0 dams, and the F1 pups are exposed the combined early life stress of depressed maternal care and the conflict between the dam and intruders. F2 juveniles and adults exhibit deficits in social behavior [17,18], and F2 dams display pervasive deficits in maternal care throughout lactation [19]. We have also established that intranasal vasopressin and oxytocin treatments in maternal rats, hormones critically involved in the regulation of social behavior and the stress response, can alter basal interferon-γ levels and social behavior in offspring [20]. Several reports have postulated that proinflammatory cytokines can impair behavior by increasing neuroplasticity in the reward pathway through direct interactions as well as indirectly by modulating neural growth factors.

Three proinflammatory immune mediators currently implicated in mediating adverse transgenerational effects of CSS on behavior are tumor necrosis factor (TNF), interferon-γ (IFN-γ), and interleukin-6 (IL-6). Stimulation of offspring blood with lipopolysaccharide (LPS) resulted in significantly more TNF and IFN-γ secretion from male offspring separated from their mothers as infants compared to infants not separated [4]. Males with major depression and a history of early life stress display an increased IL-6 response to social stress [21]. IFN-γ is a critical mediator of the immune response [22], and this cytokine mediates hyper-connectivity of frontocortical brain regions and social behavior deficits in mice [8]. Recent multigenerational data from the CSS model suggest that the immune system could be involved in additive effects across generations which would explain some of the genetic and symptomatic heterogeneity of several disorders that involve deficits in social behavior (depression, anxiety, PTSD, bipolar, schizophrenia, autism) [19].

In the current study, we assess two aspects of the transgenerational CSS model and its effects on the immune system. First, we asked whether exposure to CSS changes basal levels of IL-6, TNF, IFN-γ, and social behavior in CSS F1 female juvenile rats to follow up on the initial study of F2 animals [18]. Second, we assessed functional immunological consequences by asking whether CSS affected the ability of F2 females to respond to vaccination using *Mycobacterium bovis* Bacillus Calmette–Guérin (BCG), used to prevent systemic tuberculosis [23]. We hypothesized that F1 female juveniles would display deficits in social behavior, elevated IL-6, TNF and IFN-γ, social behavior would be negatively correlated with basal cytokine levels, and that the immune response to BCG would be attenuated in F2s. We report several changes in the associations between social behaviors and cytokines in the F1 juvenile offspring of the CSS model. It is suggested that changes in the immune–behavior relationships in F1 juveniles indicate the early stages of immune mediated disruption of social behavior that becomes more apparent in F1 dams and the F2 generation. We also report an impaired interferon response to vaccine challenge. It is hypothesized that the effects of social stress may accumulate over generations through changes in the immune system, establishing the immune system as an effective preventative or treatment target for social behavior pathologies and impaired immune responses.

## 2. Materials and Methods

### 2.1. Chronic Social Stress (CSS) Model

All rat procedures were approved by Tufts Institutional Care and Use Committee, protocol G2015-10, approved 1/12/2015. Sprague-Dawley rats (Charles River Inc., Kingston, NY, USA) in this study were maintained in accordance with the guidelines of the Committee of the Care and Use of Laboratory Animals Resources, National Research Council. Twenty-six virgin female and eight experienced breeder male rats were purchased from Charles River Inc., Kingston, NY, USA and used to establish breeding trios (two females (F0 generation) and one male per cage housed together for 4 days a week) over a period of 4 weeks. Briefly, F0 dams (which refers to pregnant F0 females) were divided randomly into control (*N* = 13 F0 dams) or CSS (*N* = 13 F0 dams) groups. Following parturition, F0 dams were singly housed with their litters of F1 pups. F0 dams and their F1 pups in the CSS group were exposed to CSS throughout lactation by placing an unknown intruder male of similar size into the home cage of the female and pups for one hour each day between days 2 and 16 of lactation. A typical social stress interaction consists of the male approaching the dam and litter which induces maternal aggression from the dam towards the male (bodily contact, kicking, biting) and elicits defensive behaviors from the male (fending off with forelimbs, crouching, retreat). Video of CSS behavioral interaction can be found online [13]. This represents a combined early life stress of depressed maternal care and exposure to the maternal-intruder conflict for the F1 pups [10,11,12], which were continuously monitored during male intrusion to ensure pup safety. Control F0 dams and their F1 pups were not exposed to daily intrusion stress. 

The F1 and F2 samples involved one pup from F0 and F1 litters; there were not multiple pups from F0 or F1 litters, and each litter was housed in a single cage. One female F1 pup per litter from the F0 control and CSS groups was weaned at day 21, raised to adulthood, and mated with a breeder male to generate F2 offspring. Analysis of maternal care in F0, F1 and F2 generations reveals an accumulation of deficits in maternal care across generations [19]. Final samples sizes were 9 female juveniles for both the control and CSS groups in the F1 generation (based on F0 mating success during the experimental period) and F2 female sample sizes of 6 for control and 3 for CSS. There were no differences in number of pups or litter weights between the F1 and F2 control and CSS litters at birth (all *p* values > 0.2), and litters were culled to five males and five females on postnatal day 1.

### 2.2. Behavioral Analyses

After weaning at 21 days of age, F1 juvenile females were tested for social behavior between days 35–40. Juveniles were placed alone in a 12" × 20" × 12" black acrylic open field box with a ceiling mounted video camera. The open field was additionally fitted with two full-length dividers, one an opaque black acrylic and the other a transparent clear acrylic. Two F1 juveniles of the same group (Control or CSS) and gender were placed into the divided open field, one in each section. One of the juveniles was marked with a line of sharpie on their back to help distinguish them on film. The rats were then left alone in the room. After five minutes of acclimation, the opaque divider was replaced with the transparent divider, and the rats were again left in the room. After ten minutes, the transparent divider was removed and the rats were allowed to interact physically for fifteen minutes and video recorded. After testing, the rats were placed back into their separate home cages, the walls and floor of the open field were disinfected with chlorhexidine solution, and the next pair of rats was tested. Behaviors scored consisted of self grooming, anogenital (AG) investigation, chasing, flight (running away from chasing juvenile), and allogrooming. The videos were analyzed using Odlog (Macropod Software) to determine total durations and frequencies of behaviors during the 15 min direct social interaction period. 

### 2.3. Vaccination with M. Bovis BCG

*M. bovis* BCG was obtained from BEI Resources and grown to mid-log phase in supplemented 7H9 broth as described [24]. Female F2 control and CSS rats were vaccinated once subcutaneously in the flank with 1 × 10^6^ Colony Forming Units (CFU) of BCG in 200 μL of sterile PBS at 8 weeks old. Eight weeks following vaccination (16 weeks of age), F2 females were euthanized and the spleens collected. The timing, dose, and site of vaccination are based on standard practices in mouse models [24].

### 2.4. Cytokine Measurements from F1 Blood and F2 Spleen Cultures

Following euthanasia of F1 females, heparinized trunk blood was diluted 1 to 5 in complete cell culture media as described [25,26]. After 48 h at 37 °C 5% CO2, blood cultures were frozen at −20 °C until use. Following euthanasia of BCG vaccinated CSS and Control F2, single cell suspensions from the spleen were cultured with and without *M. bovis* antigens at a final concentration of 5 μg/mL, media alone (negative control) or concanavalin-A (Con A, positive control) modified from [25,26]. After 48 h at 37 °C 5% CO2, spleen cultures were frozen at −20 °C until use. For cytokine quantification, blood and spleen cultures were thawed, and IL-6, IFN-γ and TNF were quantified using commercially available, validated, Quantikine ELISA kits from R&D Systems (Minneapolis, MN, USA), following the manufacturer’s directions. Experimental and assay positive and negative controls worked as expected.

### 2.5. Statistical Analyses

Litter effects were avoided by the use of 1 pup per F1 and F2 litter (Table 1). Behavior durations and frequencies and cytokine concentrations in the control and CSS groups were compared with unpaired 2-tailed *t*-tests to determine effects of CSS. Pearson correlations were used to test for significant cytokine/behavior associations on the combined control and CSS data, as well as each treatment group independently. 1-tailed tests were used if initial 2-tailed correlations were identified with a particular cytokine. Statistical significance was denoted as *p* < 0.05.

## 3. Results

### 3.1. F1 Cytokine Concentrations and Behavior

There were no effects of CSS on F1 female juvenile basal plasma IL-6, TNF, or IFN-γ levels (Table 2). There were no effects of CSS on F1 female juvenile behavior during a social behavior test (Table 3).

### 3.2. F1 Behavior–Cytokine Correlations

Self grooming duration (*p* = 0.03, Figure 1A) and frequency (*p* = 0.04, Figure 1B) were negatively correlated with IL-6 in control F1 female juveniles, but not CSS female juveniles. AG investigation duration was positively correlated with IL-6 in CSS female juveniles (*p* = 0.03, 1-tailed *t*-test), but not control females (Figure 1C). Chase frequency was negatively correlated with IL-6 in both groups combined (*p* = 0.04, Figure 1D).

AG duration (*p* < 0.05, 1 tailed *t*-test, Figure 2A) and frequency (*p* = 0.01, Figure 2B) were positively correlated with TNF in control F1 female juveniles only. Self grooming duration was negatively correlated with IFN in both groups combined (*p* = 0.02, Figure 3).

### 3.3. F2 Cytokine Production Following Vaccination

The spleen cultures from F2 CSS females produced significantly less IFN-γ in response to BCG antigens and there was a general trend for diminished IFN-γ production in response to concanavalin-A than F2 rats with no exposure to CSS (Figure 4). There was a trend for increased IL-6 in response to BCG antigens and in response to concanavalin-A (*p* = 0.08). No clear trend was observed with TNF. In all cases, there were no significant differences in unstimulated cultures.

## 4. Discussion

Although there were no significant effects of early life CSS on basal cytokine levels or social behavior, regression analyses revealed several changes in the relationships between IL-6, TNF and IFN and anxiety and social behaviors. Taken in the context of the related CSS studies of F1 dams and F2 juveniles, adults, and dams, it is postulated that these disrupted cytokine behavior relationships in F1 juveniles represent the early stages of adverse transgenerational effects of CSS on social behavior. The F2 cytokine results indicate that CSS may have transgenerational consequences on both acquired and innate immune responses, possibly by maintaining a generalized state of inflammation involving IL-6 while simultaneously reducing the capacity to develop effector and memory acquired immune responses mediated by IFN-γ, which are critical mechanisms by which vaccines provide durable immunity.

The lack of effects of CSS on F1 juvenile behavior confirms previous unpublished negative data from F1 juveniles. In contrast, F1 dams express depressed maternal care and elevated maternal anxiety during maternal care testing [12]. The presence of substantial behavioral effects of CSS in F1 dams, but not female F1 juveniles, could be explained by the robust neuroendocrine changes during pregnancy, parturition, and lactation [27] which accentuate non-significant effects at the juvenile stage. Data from humans suggests that IL-6 levels are higher in parents compared to singles and couples [28], and behavior/immune associations might be more substantial during periods of intense bonding, as with parental—offspring bonding compared to the juvenile social interactions of the current study. While there is a negative association between IL-6 and self grooming in the control F1 juveniles, this relationship may be either generally disrupted in CSS animals and/or transitioning to a positive association where high basal levels of IL-6 mediate anxiety associated increased levels of self grooming, as IL-6 has been specifically linked to anxiety [29] and susceptibility to social stress [7]. IL-6 may represent a useful biomarker for stress induced depression and anxiety disorders, and manipulations of this cytokine could have potent preventative or ameliorative effects.

With AG investigation and IL-6, the relationship was different between the two treatment groups in that AG investigation and IL-6 were positively correlated only in the CSS exposed animals. This may indicate the development of a transgenerational mechanism for the increased investigatory behavior and decreased allogrooming in CSS F2 females [17], which is interpreted as deficient social recognition due to the lack of typical progression from social recognition to direct social interaction (allogrooming). The overall negative association between IL-6 and chase frequency suggests that IL-6 is not likely to be involved in CSS induced changes in chase behavior, but may be generally involved in the modulation of this aspect of social investigation. Based on recent investigations of the role of IL-6 in social stress susceptibility [7,30,31], the trend of increased IL-6 in vaccinated rats may represent a biomarker for increased susceptibility of F2 females to IL-6 related social stress pathologies, such as depressed maternal care [3]. While further studies are warranted to confirm how CSS alters acquired immune responses generated by BCG vaccination, the combination of the F1 and F2 IL-6 data supports previous findings on the role of this cytokine in social stress associated depression and anxiety and indicates that further research on this topic is warranted.

In contrast to the IL-6/AG data, TNF is positively correlated with AG duration and frequency in controls only, indicating that the cytokine/behavior relationship changes can be cytokine specific. Again, the lack of association in the CSS juveniles could represent variation in the susceptibility to the effects of CSS on social behavior, with high levels of AG investigation being induced by TNF only in susceptible animals. Cancer patients reporting increased social activities and social satisfaction display enhanced stimulated TNF responses [32], and this cytokine is sensitive to social stressors [33]. The presence of associations between both IL-6 and TNF and investigatory behavior suggest that interactions between these two cytokines may mediate initial social investigation, possibly through changes in anxiety and aggression related responses to social interaction [34,35].

The negative IFN-γ self grooming association suggests that this immune factor is generally involved in the regulation of self grooming with low IFN-γ values being associated with high levels of self grooming. Previous CSS studies have not found significant associations between IFN-γ and behavior in female F2 juveniles [20], and one explanation is that IFN-γ’s effects are more involved in overall neuronal development than the modulation of specific behaviors, and/or that downstream growth or immune factors are more critical to the expression of social behaviors. Previous studies of IFN-γ and social stress indicate that it is elevated in aggressive male mice [35]. However, while rearing in isolation also increases IFN-γ in male mice [31] and this cytokine is necessary for the expression of depressive behavior in BCG infected male mice [36,37], there is no effect of isolated rearing in females, suggesting important sex differences in the immune response to social stress. A role for sex differences is supported by the basal immune and behavioral data from CSS F2 animals [18].

Another avenue to explore is the role of CSS-induced immune changes that have far-reaching transgenerational consequences on protection induced by common childhood vaccinations. In the current study, we demonstrate that CSS significantly reduces antigen-specific responses to BCG, the most widely used vaccine in human children across the globe, to prevent childhood manifestations of TB: high mortality systemic disease and meningitis. This preliminary work (due to small sample sizes) implies that BCG vaccination is less effective under conditions of transgenerational stress and could also indicate that CSS F2 animals exhibit impaired responses to infections independent of vaccination, especially infections that require IFN-γ mediated activation of innate immunity to clear or control infection or carcinogenesis. Chronic psychosocial stress increases the risk for colon carcinogenesis in male mice and may suppress colonic IFN-γ activity [38]. CSS alters both basal [18] and infection related levels of immune factors in F2 offspring. Earlier studies of F2 males and females reported decreased basal corticosterone levels at the juvenile stage [17] which may mediate immune changes. In humans, neonatal dexamethasone treatment results in increased IFN-γ production in the children at 7–10 years [39]. It is postulated that corticosteroid levels during early life in CSS F2 females programmed the immune response system, resulting in the attenuated IFN-γ response to vaccine challenge. The current data and related studies raise important questions about bidirectional interactions between stress, behavior, the social environment, poverty, and infectious diseases.

## 5. Conclusions

Transgenerational social stress alters immune–behavior associations and may impair responses to vaccination. In the context of related CSS studies, the current data support transgenerational effects of social stress on behavior and response to vaccination. There is growing evidence of the genetic, etiological, and symptomatic heterogeneity of disorders that have maladaptive effects on social behavior (depression, anxiety, bipolar, schizophrenia, PTSD, autism). Accumulating transgenerational immune-based mechanisms may explain some of this vast diversity [19]. It is suggested that future studies of the biological basis of social behavior carefully integrate transgenerational and infection related facets using ethologically, etiologically, and translationally relevant social and immune manipulations to generate novel developmental and mechanistic conclusions. Investigations of comorbidities between immunological disorders and treatments, alterations in social behavior, and related neural mechanisms may be especially productive with identifying innovative preventative and treatment strategies for social deficits.

## Figures and Tables

**Figure 1 brainsci-07-00089-f001:**
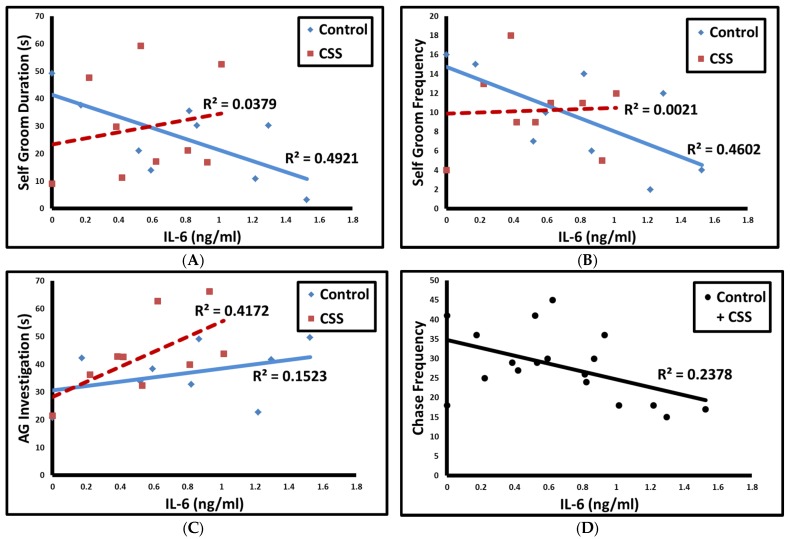
Correlations between IL-6 and self grooming duration (**A**); self grooming frequency (**B**); AG investigation duration (**C**); and chase frequency (**D**). Solid blue line = control, dashed red line = CSS, solid black line = both groups combined for chase frequency.

**Figure 2 brainsci-07-00089-f002:**
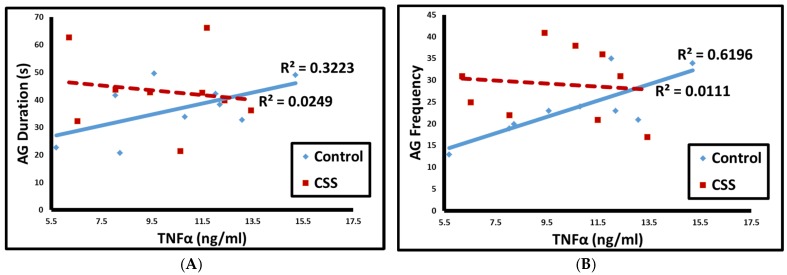
Correlations between TNF and AG investigation duration (**A**) and frequency (**B**). Solid blue line = control, dashed red line = CSS.

**Figure 3 brainsci-07-00089-f003:**
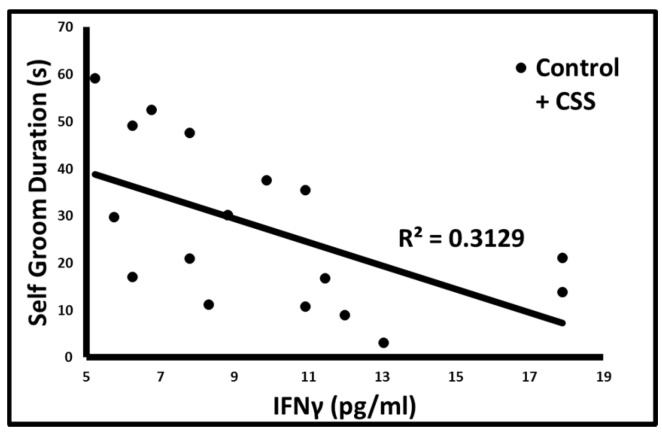
Correlation between IFN and self grooming duration in both groups combined.

**Figure 4 brainsci-07-00089-f004:**
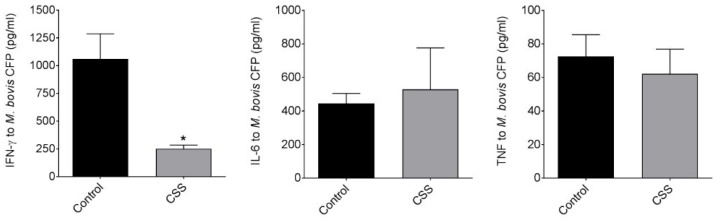
Differential effects of transgenerational stress on acquired and innate immune responses at the F2 generation. F2 rats were vaccinated with 1 × 10^6^ CFU of *M. bovis* BCG subcutaneously, euthanized 8 weeks later, and splenocytes cultured with *M. bovis* culture filtrate protein (CFP) to assess antigen-specific T cell responses or with Concanavalin-A as a strong non-specific stimulant. The cultures from F2 CSS females produced less IFN-γ in response to BCG-specific stimulation and to non-specific stimulation with Concanavalin-A than F2 rats with no exposure to CSS. In contrast, a trend for increased IL-6 production was observed under identical conditions in the same cultures. There was no clear relationship between CSS exposure in the F2 generation and TNF production.

**Table 1 brainsci-07-00089-t001:** Summary of experimental procedures.

**F1 Females**	***n***	**Tests**
Control	9	Social Behavior/Plasma Cytokines
CSS	9	Social Behavior/Plasma Cytokines
**F2 Females**		
Control	6	BCG vaccination/Spleen Cytokine responses to BCG and ConA
CSS	3	BCG vaccination/Spleen Cytokine responses to BCG and ConA

**Table 2 brainsci-07-00089-t002:** Mean ± SEM basal plasma cytokine values from control and CSS F1 juvenile females with *p*-value results from 2-tailed *t*-tests.

Cytokine	Control	CSS	*p*-Value
IL-6	0.78 ± 0.17	0.55 ± 0.11	0.27
TNF	10.5 ± 1.0	10.0 ± 0.9	0.67
IFNγ	10.5 ± 1.1	9.0 ± 1.4	0.43

**Table 3 brainsci-07-00089-t003:** Mean ± SEM durations (s) and frequencies (#) of behaviors displayed by control and CSS F1 juvenile females during a 15 min social behavior test with *p*-value results from 2-tailed *t*-tests.

Behavior	Control	CSS	*p*-Value
Self Grooming (s)	25.8 ± 4.9	29.4 ± 6.3	0.65
Self Grooming #	9.6 ± 1.7	10.2 ± 1.4	0.77
AG Investigation (s)	36.8 ± 3.4	43.1 ± 4.7	0.29
AG Investigation #	23.6 ± 2.3	29.1 ± 2.8	0.14
Chase (s)	38.0 ± 11.9	47.2 ± 7.2	0.52
Chase #	25.4 ± 3.1	30.1 ± 2.8	0.23
Flight (s)	32.7 ± 11.3	35.4 ± 6.3	0.84
Flight #	22.9 ± 3.4	28.2 ± 3.3	0.28
Allogrooming (s)	41.3 ± 9.7	42.4 ± 7.6	0.92
Allogrooming #	33.1 ± 4.4	34.6 ± 4.4	0.83

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
