# Peer review of "Transgenerational Social Stress Alters Immune–Behavior Associations and the Response to Vaccination"

_brainsci, 2017, doi:10.3390/brainsci7070089_

Round 1

Reviewer 1 Report

An interesting and important publication

Author Response

Thank you for your support!

Ben

Reviewer 2 Report

This is a interesting and novel paper. While the authors do not see huge changes induced by the stress they impose I still think this is a well written and put together manuscript.

My specific comments:

Introduction

well written but should be reduced to one page

Line 63-please expand/clarify what "behaviourally active neuropeptides" are

Line 73-"stimulation of blood..." what generation/group are they referring to?

Line 75-typo-replace composed with compared

Be concise with hypothesis and justify use of model etc. in the discussion instead

Methods

Well described but would benefit from a table to explain stressors/tests and samples collected from the different generations. Please include group numbers here. As currently it appears there may be 3 in some groups but it is not fully clear. If so this is too low.

Please justify the timing and dose of the vaccination

Why were offspring were not used?

Also justify the statistics used

Results

Mostly clearly written

Is ECSS a typo? It is not described what this group is

Discussion

Good but needs to be reduced to a page and a half-there is not a huge amount of results to discuss so tightening up the discussion would make improve the overall impression of the manuscript

Author Response

Thank you for your useful comments.  We have made the requested changes which are tracked in the revised manuscript and detailed below.

 The introduction has been shortened to one page.

 The additional minor revisions have been completed, and we are more concise with out hypotheses.

 We have added a table of experimental procedures, with sample sizes.  We agree that the vaccine work sample sizes are too low, and have qualified statements regarding these data in the abstract and discussion by noting their preliminary nature.  It is our hope that the current data stimulate more extensive studies in this area.

The timing and dose of the vaccination has been justified based on similar mouse experiments conducted by Dr. Beamer.

We did use the F1 and F2 offspring.  If the question was about male offspring, we were limited by time and financial resources, and the primary focus of the Nephew lab is on females due to the relative lack of data on female rodents.

We used the described statistics based on the comparison of two experimental groups.  The use of more complicated statistical models is not necessary due to the use one pup per litter to avoid litter effects across or within generations.

 ECSS was a typo.

The discussion has been shortened to a page and a half.

We are unable to insert and revise the citations due to the current format provided by Brain Sciences.  Please let us know if we need to make changes to assist with correcting the references and bibliography.

Thanks,

Ben